# The Prevalence and Impact of Nutritional Risk and Malnutrition in Gastrointestinal Surgical Oncology Patients: A Prospective, Observational, Multicenter, and Exploratory Study

**DOI:** 10.3390/nu15143283

**Published:** 2023-07-24

**Authors:** Manuel Durán Poveda, Alejandro Suárez-de-la-Rica, Emilia Cancer Minchot, Julia Ocón Bretón, Andrés Sánchez Pernaute, Gil Rodríguez Caravaca

**Affiliations:** 1Department of General and Digestive Surgery, Hospital Universitario Rey Juan Carlos, 28933 Madrid, Spain; 2Department of Medical Specialties and Public Health, Faculty of Health Sciences, Rey Juan Carlos University, 28933 Madrid, Spain; gil.rodriguez@salud.madrid.org; 3Department of Anesthesiology and Surgical Critical Care, Hospital Universitario La Princesa, 28006 Madrid, Spain; alejandro.suarez.delarica@gmail.com; 4Department of Endocrinology and Nutrition, Hospital Universitario de Fuenlabrada, 28942 Madrid, Spain; emilia.cancer@salud.madrid.org; 5Department of Endocrinology and Nutrition, Hospital Universitario “Lozano Blesa”, 50009 Zaragoza, Spain; mjocon@salud.aragon.es; 6Department of General and Digestive Surgery, Hospital Clínico San Carlos, 28040 Madrid, Spain; pernaute@yahoo.es; 7Department of Preventive Medicine, Hospital Universitario Fundación Alcorcón, 28922 Madrid, Spain

**Keywords:** malnutrition, digestive surgery, oncology, gastrointestinal malignancies, prevalence

## Abstract

A prospective, observational, multicenter, and exploratory study was conducted in 469 gastrointestinal cancer patients undergoing elective surgery. The Malnutrition Universal Screening Tool (MUST) and the Global Leadership Initiative on Malnutrition (GLIM) criteria were used to assess nutritional risk. On admission, 17.9% and 21.1% of patients were at moderate (MUST score 1) and severe (MUST score ≥ 2) nutritional risk, respectively. The GLIM criteria used in patients with a MUST score ≥ 2 showed moderate malnutrition in 35.3% of patients and severe in 64.6%. Forty-seven percent of patients with a MUST score ≥ 2 on admission had the same score at discharge, and 20.7% with a MUST score 0 had moderate/severe risk at discharge. Small bowel, esophageal, and gastric cancer and diabetes were predictors of malnutrition on admission. Complications were significantly higher among patients with a MUST score 1 or ≥2 either on admission (*p* = 0.001) or at discharge (*p* < 0.0001). In patients who received nutritional therapy (*n* = 231), 43% continued to have moderate/severe nutritional risk on discharge, and 54% of those with MUST ≥ 2 on admission maintained this score at discharge. In gastrointestinal cancer patients undergoing elective surgery, there is an urgent need for improving nutritional risk screening before and after surgery, as well as improving nutritional therapy during hospitalization.

## 1. Introduction

It has been extensively recognized that malnutrition in hospitalized patients negatively affects quality of life and prognosis by increasing morbidity, mortality, length of hospital stay, reducing the response to treatment (active treatment including surgery), and increasing both re-admission rates and health care costs [1,2,3,4,5,6]. The prevalence of malnutrition in the hospital setting ranges between 20% and 50% [7,8,9]. In a nationwide cross-sectional multicenter study carried out in Spanish hospitals, almost one in four patients was malnourished [10]. Also, malnutrition was associated with an increase in length of hospitalization, especially in patients admitted without malnutrition, but who presented malnutrition at discharge [10]. The causes of malnutrition in hospitalized patients are multifactorial, including the patient’s illness itself (disease-related malnutrition), which can interfere with the adequate absorption and metabolism of food, loss of appetite, fasting for diagnostic procedures, conditions that compromise the regular function of the digestive system, and poor management of patient nutrition [11].

In surgical patients, particularly those undergoing gastrointestinal surgery, malnutrition is one of the main comorbidities and an independent predictor of poor postoperative outcomes [12,13]. Surgery imposes further physiological stress with metabolic demands and hypercatabolism, postoperative fasting, prolonged ileus, gastric atony, malabsorption syndrome, fistula, or intestinal obstruction as frequently reported surgery-related causes of malnutrition [14]. It has been shown that as many as two of every three major surgery patients are malnourished preoperatively, a diagnosis rarely made and treated even less frequently [12]. Data from the National Surgical Quality Improvement Program (NSQIP) demonstrate that malnutrition is one of the only major readily modifiable preoperative risk factors associated with poor surgical outcomes, including mortality [15].

In cancer patients, malnutrition is present at the initial diagnosis in about 15–40% of cases and this incidence increases during active oncologic treatment (radiotherapy, chemotherapy, immunotherapy, and targeted agent) affecting 40–80% of patients, particularly in those with advanced disease [16]. Malnutrition has a definitive impact on several aspects of cancer treatment and outcome, reducing dose intensity and treatment response, increasing treatment toxicities, and worsening the patient’s quality of life and functionality, with weight loss and loss of skeletal muscle mass as two hallmarks of cachexia in advanced cancer patients [17]. Despite strong recommendations from clinical practice guidelines regarding the integration of nutrition into the overall management of surgical and cancer patients [18,19,20], as well as awareness of surgeons that nutrition intervention can reduce postoperative complications and improve outcome [21,22], perioperative nutrition practices continue to be suboptimal in daily clinical routine [23,24,25].

Moreover, beyond the stage and the treatment received, malnutrition is consistently associated with specific cancer type, such as oral, lung, and gastrointestinal cancer. Studies in colorectal cancer patients have reported prevalence rates of preoperative malnutrition of up to 40% [26,27,28,29]. In a prospective cohort study of 649 surgical cancer patients admitted to the intensive care unit (ICU) after major abdominal surgery, the prevalence of malnutrition before surgery was 65.3% [14]. In the NutrCancer2021 study, a prevalence of malnutrition in gastrointestinal oncological patients on hospital admission of approximately 50% was reported (liver cancer 55%, pancreatic cancer 54%, and upper gastrointestinal cancer 53%) [30].

The present study was conducted to determine the prevalence and impact on outcome of nutritional risk and malnutrition at admission to the hospital and at hospital discharge in adult patients undergoing elective major abdominal surgical procedures for the treatment of gastrointestinal cancer.

## 2. Materials and Methods

### 2.1. Study Design and Patient Population

This was a nationwide, prospective, observational, multicenter, and exploratory study (the PREMAS Study, “*PRE*valence of *MA*lnutrition in gastrointestinal Surgical oncology patients”). The study was carried out in Spain, with the participation of public hospitals, in which major abdominal surgical procedures in cancer patients are routinely performed. At least one center, with 300 or more beds and located in a major Spanish region, participated in the study. The primary objective of the study was to determine the prevalence of nutritional risk using the Malnutrition Universal Screening Tool (MUST) [31] and the prevalence of malnutrition using to the Global Leadership Initiative on Malnutrition (GLIM) criteria [32] in adult oncological patients at the time of admission to the hospital for elective gastrointestinal surgical procedures. Secondary objectives included: (1) to describe the course of nutritional status from hospital admission to discharge; (2) to identify predictors of malnutrition at the time of admission to the hospital and at hospital discharge; and (3) to assess the impact of nutritional risk at hospital admission on complications and the effect of nutritional therapy during hospitalization on nutritional risk at hospital discharge.

Between July 2020 and May 2021, male and female patients aged 18 years or older diagnosed with a malignant neoplasm of the gastrointestinal tract who had been scheduled to undergo a major abdominal surgical procedure in the participating hospitals were eligible. Inclusion criteria were to be admitted to the hospital up to two days before the operation, an expected length of stay of at least 5 days postoperatively, and to provide written informed consent. Exclusion criteria were as follows: age under 18 years, minor or non-elective surgical procedures as the reason for hospital admission, presence of secondary malignant tumors of the gastrointestinal tract, patients wearing temporary or palliative intestinal prosthesis, concurrent participation in another interventional study, and refusal to sign the informed consent form.

The study protocol was approved by the Ethics Committee for Clinical Research (CEIC) of Hospital Clínico San Carlos (Madrid, Spain) (code 20/121-E, approval date 25 February 2020) and was conducted in accordance with principles of the Declaration of Helsinki. Written informed consent was obtained from all participants.

### 2.2. Procedures and Data Collection

Screening for the risk of malnutrition was performed within 48 h after hospital admission using MUST, developed by the British Association for Parenteral and Enteral Nutrition (BAPEN) [31]. MUST is a five-step screening tool that identifies adults at risk of malnutrition based on the measurement of body mass index (BMI), unplanned weight loss in the past 3–6 months, and the effect of acute disease. The overall risk of malnutrition is classified as score 0: low risk, score 1: medium risk (malnourished), and score ≥ 2: high risk. The prevalence of nutritional risk was estimated as the percentage of patients with a MUST score ≥ 2. Moderate and severe malnutrition according to the GLIM criteria [32] was calculated in the subset of patients with a MUST score ≥ 2. Moderate malnutrition was defined as unintended weight loss of 5–10% <6 months or 10–20% >6 months, low BMI < 20 kg/m^2^ if <70 years or <22 kg/m^2^ if >70 years, and mild-to-moderate reduced muscle mass. Severe malnutrition was defined as unintended weight loss of >10% <6 months or >20% >6 months, low BMI < 18.5 kg/m^2^ if <70 years or <20 kg/m^2^ if >70 years, and severe reduced muscle mass. Nutritional assessment using the MUST and GLIM tools was also evaluated at the time of hospital discharge.

In all patients, the following data were recorded: age; gender; race; civil status; living conditions; educational level; working status; place of residence; smoking habits; weight; height; BMI; location of gastrointestinal cancer; TNM classification; type of surgical procedure; comorbidities; the Charlson comorbidity index [33]; the Barthel index [34]; nutritional treatment during hospitalization; concomitant oncological treatment before admission; inclusion in The Enhancement Recovery After Surgery Program (ERAS) of the Spanish Group of Multimodal Rehabilitation (GERM) [35]; and length of stay in the hospital and in the intensive care unit (ICU).

The Charlson comorbidity index can predict short- and long-term outcomes (risk of death attributable to comorbid disease) based on a list of 19 medical conditions, with a weight assigned to each condition from 1 to 6, and a total score ranging from 0 to 37. The Barthel index includes 10 activities of daily living to assess functional independence, with 0 to 5 points per item, and a total score from 0 to 100 (scores 0–20 indicate total dependency, 21–60 severe dependency, 61–90 moderate dependency, 91–99 slight dependency, and 100 total independency).

### 2.3. Statistical Analysis

The sample size was calculated according to a prevalence of malnutrition in gastrointestinal oncological patients on hospital admission of approximately 50% reported in the NutrCancer2021 study [30]. Therefore, considering a prevalence of malnutrition of 50% and 95% confidence interval (CI), a 5% precision and a 14% patient abandonment rate, a minimum of 450 patients should be included in the study. Based on an expected participation of about 25 hospitals throughout Spain, between 12 and 24 patients recruited per hospital would be required.

Categorical data are expressed as frequencies and percentages, and continuous data as mean and standard deviation (SD) with 95% confidence intervals (CI). The percentage of patients with a MUST score ≥ 2 at admission to the hospital and at hospital discharge was compared with the Fisher’s exact test. Also, the distribution of study variables according to the presence or absence of malnutrition (a MUST score ≥ 1 vs. a score of 0) was compared with the Fisher’s exact test for categorical variables, and the Student’s *t* test, the Mann–Whitney *U* test, or the Kruskal–Wallis test for quantitative variables according to conditions of application. Statistical significance was set at *p* ≤ 0.05. Variables with a *p* < 0.2 in the bivariate analysis were included in a stepwise logistic regression model to assess independent predictors of malnutrition both at the time of admission to the hospital and at hospital discharge. The odds ratio (OR) and 95% CI were calculated. The Statistical Analysis System (SAS Institute. Cary, NC, USA) version 9.4 was used for the analysis of data.

## 3. Results

### 3.1. General Characteristics of the Patients

Twenty-three hospitals participated in the study and a total of 469 patients were recruited and analyzed. Sixty-two percent of patients were men, aged between 23 and 93 years. The majority of patients (98.9%) were Caucasian. Seventy-one percent (*n* = 333) of patients were married or had a partner and 78.2% (*n* = 367) lived with his/her partner or their own family. Also, more than half of the patients (56.3%) had a primary education, 11.1% had university degrees, 60.5% were retired, and 69.55 lived in cities of more than 50,000 inhabitants. Salient anthropometric and clinical data are shown in Table 1.

The mean (SD) age of the patients was 68.2 (11.7) years, with a mean BMI of 26.7 (4.3) kg/m^2^ (36% of patients had a BMI < 24.9 kg/m^2^). In relation to body weight in the previous 6 months, 65.5% of patients reported a percentage of weight loss of less than 5% and 34.3% a percentage of weight loss between 5% and 30%. In the group of 70 active smokers, the mean number of pack years was 30.2 (28.1). The mean Charlson comorbidity index was 2.86 (1.81), with a short-term (<3 years) risk of death attributable to comorbid disease of 26% in 43% of the patients, and a long-term (>5 years) risk of death attributable to comorbid disease of 85% in 63% of the patients. On the other hand, 82.5% of the patients were totally independent, with a 100 score in the Barthel index.

### 3.2. Gastrointestinal Cancer and Perioperative Data

As shown in Table 2, colon cancer was the most frequent primary tumor (53.9% of patients) followed by cancer of the rectum (23.7%) and gastric cancer (8.1%). In relation to the surgical procedures, right hemicolectomy, sigmoidectomy, low anterior rectal resection, left hemicolectomy, and gastrectomy were the most common surgical techniques.

Of the 158 patients on nutritional support at the time of admission to the hospital, 154 (97.5%) received nutritional supplements. Nutritional support was more frequent among patients included in an ERAS program (*n* = 95, 60.1%) as compared to those not included in an ERAS program (*n* = 63, 39.9%) (*p* < 0.001).

### 3.3. Primary Objective: Prevalence of Nutritional Risk and Malnutrition on Admission

At the time of admission to the hospital, the mean overall score of malnutrition established by MUST was 0.72 (1.09) (95% CI 0.62–0.82). Ninety-nine patients (21.1%) were at high risk of malnutrition with a MUST score ≥ 2. Moderate nutritional risk (MUST score 1) was present in 84 patients (17.9%). In most patients (61.0%), there was no risk of malnutrition with a MUST score of 0 (Table 3). In the group of 99 patients at risk of severe malnutrition (MUST score ≥ 2), 35 patients (35.3%) met the GLIM criteria of moderate malnutrition and 64 (64.6%) of severe malnutrition. Therefore, the risk of severe malnutrition was 21.1% according to the MUST score and from those 64.6% according to the GLIM criteria (Table 3).

### 3.4. First Secondary Objective: Course of Nutritional Status from Admission to Discharge

At the time of discharge from the hospital, data from five patients were missing. The mean MUST score was 0.55 (0.93) (95% CI 0.46–0.63). As shown in Table 4, there were 65 (14.0%) patients at severe nutritional risk (MUST score ≥ 2), 93 (20.0%) at moderate risk (MUST score 1), and 306 (66.0%) without nutritional risk (MUST score 0). Differences in MUST scores between admission and discharge were statistically significant (*p* < 0.001). This significant differences was related to fewer patients at nutritional risk at discharge (66% [306/464]) compared to admission (61% [289/469]).

On admission to hospital, the percentage of patients at nutritional risk (MUST score 1 or ≥2) was 39.0% (183/469) and decreased to 34.0% (158/464) at the time of hospital discharge. The percentage of patients with severe malnutrition risk (MUST score ≥ 2) decreased from 21.1% to 13.9%. However, 20.7% (59/285) of patients without nutritional risk on admission showed moderate/severe risk at discharge. A total of 47.4% (45/95) patients with MUST score ≥ 2 on admission remained at severe risk at the time of discharge. Also, 38.9% (37/95) patients who were at severe nutritional risk on admission (MUST score ≥ 2) had no risk (MUST score 0) at discharge from the hospital.

At the time of hospital discharge, 65 patients (14.0%) were at risk of severe malnutrition according to the MUST score, and according to the GLIM score, moderate malnutrition was present in 14 patients (21.5%) and severe malnutrition in 51 (78.5%).

### 3.5. Second Secondary Objective: Predictors of Malnutrition at Hospital Admission and Discharge

In relation to predictors of malnutrition (MUST score ≥ 1) on admission to the hospital, variables with a difference of *p* < 0.20 between the categories of MUST 0 and MUST ≥ 1 in the bivariate analysis are shown in Table 5.

In the multivariate analysis, variables independently associated with the risk of malnutrition (MUST score ≥ 1) were small bowel, gastric, and esophageal cancer as well as diabetes mellitus (Table 6).

In relation to the prediction of malnutrition (MUST score ≥ 1) at hospital discharge, variables with a *p* value ≤ 0.02 between the groups of MUST score 0 and ≥1 in the bivariate analysis are shown in Table 7.

In the logistic regression analysis, the risk of moderate and severe malnutrition (MUST ≥ 1) at hospital discharge is statistically correlated with the risk of malnutrition at admission, the type of surgery with the digestive cytoreduction, the nutritional intervention during hospitalization, and the occurrence of postoperative complications during hospital stay (Table 8).

### 3.6. Third Secondary Objective: Impact of Nutritional Risk on Complications and Effect of Nutritional Therapy on Nutritional Risk at Discharge

The presence of complications was statistically significantly more frequent among patients with malnutrition (MUST score 1 or ≥2) at hospital admission (Table 9). In 283 patients with a MUST score of 0 on admission, the rate of complications was 28.6% (81/283) as compared with 36.3% (65/179) in those with MUST scores of 1 or ≥2 (*p* = 0.001). Also, the rate of complications was significantly higher in patients with malnutrition at hospital discharge (MUST score 1 or ≥2) (44.9% [71/158]) than in those with no risk of malnutrition (24.7% [75/304]) (*p* < 0.0001) (Table 9).

A total of 231 patients (49.2%) received nutritional therapy during their stay in the hospital, including oral nutritional supplements in 57.3% of cases, total parenteral nutrition through the central line in 38.7%, peripheral parenteral nutrition in 24.9%, and enteral nutrition through a nasojejunal feeding tube or jejunostomy in 4.9%.

In patients who received nutrition support, there were statistically significant differences between MUST scores on admission and at hospital discharge (*p* < 0.0001). As shown in Table 10, 43.7% (101/231) of patients continued to present moderate or severe risk of malnutrition (MUST score 1 or ≥2) at hospital discharge, and 54.3% (38/70) of patients with MUST score ≥ 2 on admission maintained the same score at discharge. Moreover, among the 38 (54.3%) patients with a MUST score ≥ 2 at hospital discharge treated with nutritional support, 77.6% (38/49) of them continued to present severe malnutrition.

In relation to changes of the GLIM score, 41.7% of patients with moderate malnutrition on admission had severe malnutrition at discharge, whereas 3.8% with severe malnutrition on admission had moderate malnutrition at discharge (*p* = 0.0004). However, 78.9% of patients with severe malnutrition maintained this nutritional status at discharge (Table 11).

## 4. Discussion

This study carried out on a large sample of adult patients with primary gastrointestinal malignancies undergoing elective major abdominal procedures shows that, at the time of admission to the hospital, 21.1% were at severe risk of malnutrition based on a MUST score ≥ 2. The overall percentage of patients at any risk of malnutrition (MUST score ≥ 1) was 39.0%. At hospital discharge, however, the risk of severe malnutrition decreased to 13.9%, and the overall risk of malnutrition to 33.7%. The risk of malnutrition based on the GLIM criteria, assessed in patients at severe risk based on a MUST ≥ 2, was moderate in 35.3% of the patients and severe in 64.6%. However, at the time of hospital discharge, there was a decrease in the group of moderate malnutrition (from 35.3% to 21.5%), but patients with severe malnutrition increased from 64.6% to 78.5%. Despite the fact that nutritional support was provided to about 48% of patients during in-patient care, 14% of patients continued to be at high risk of malnutrition at the time of discharge from the hospital. These findings confirm that malnutrition in gastrointestinal surgical oncological patients is a clinically relevant problem and remains an unmet medical need.

The negative impact and consequences of malnutrition in the hospital setting is well known and demonstrated in particular for cancer patients. It has been emphasized in several studies showing an increase in mortality and treatment toxicity and a decrease in treatment response and patent compliance to the treatment [14,16,27,36]. In a large multicenter study on disease-related malnutrition in Spain, conducted in 17 hospitals during a period of 5 to 7 days in 2185 patients (SeDREno study), malnutrition was observed in 29.7% of the patients, a somewhat higher percentage than the 21.1% found in our study, although the percentage of severe malnutrition was 12.5%, lower than the 13.9% reported here [9]. The mean age of the patients was 67.1 years, which was similar to the 68.2 years found in our population. Diabetes was a concomitant disease in 34.8% of patients in the SeDREno study [9] and 23.4% in our study, although in both multivariate analyses, diabetes was an independent factor associated with malnutrition.

In a review of different tools to identify patients at risk of malnutrition or who were malnourished, a comparison among three nutritional screening tools (MUST, Subjective Global Assessment [SGA], and the Nutritional Risk Screening-2002 [NRS-2002]) in old hospitalized patients showed that MUST is the most sensitive, specific, and accurate system in identifying malnourished patients despite being less rapid as compared to NRS-2002 and SGA [11,37]. The MUST scores used in our study are a popular screening tool for all types of hospitalized patients and have been shown to be fast, reproducible, and easy to use [38].

In a case–control nested cohort study of elderly patients (>70 years) undergoing elective colorectal cancer (CRC) surgery and using the Controlling nutritional status (CONUT) score > 4 points to define malnutrition, the prevalence of malnutrition was 30.1% [26], which is similar to 35.3% of moderate malnutrition according to the GLIM criteria found in our study. That study showed that CRC patients with severe nutritional risk at admission had a significantly increased mortality at 1 year after discharge [26]. In our study, however, in the group of 253 patients with cancer of the colon, the risk of malnutrition at hospital admission was present in 46.4% of cases, although colon cancer was not selected as predictor of malnutrition in the regression analysis. By contrast, gastric cancer, small bowel cancer, and esophageal cancer were independently associated with the risk of malnutrition. In a study of 5309 patients who had undergone a gastric resection procedure for gastric cancer, from which there were 1044 with malnutrition and 1044 matched controls, malnutrition was associated with increased postoperative mortality, length of hospital stay, and hospitalization costs [39].

In the PREDyCES study, a nationwide, cross-sectional observational multicenter study carried out in Spanish hospitals in which the NRS-2002 screening tool for malnutrition was used, 776 patients out of 1706 (45.5%) had malnutrition on admission to hospital as compared with 369 out of 1576 (23.4%) at hospital discharge [10]. Mean hospital stay was significantly longer in malnourished patients at admission and at discharged as compared to non-malnourished patients [10]. In the multivariate analysis, low BMI (<18.5 kg/m^2^), age of 70 or over, diagnosis of neoplasia or diabetes, referral of dysphagia, and polymedication were associated with malnutrition on admission [10].

Moreover, an investigation in a large Norwegian hospital from 2008 to 2018 revealed that the prevalence of malnutrition in surgical patients of 21.2% (the same as in our study) showed no apparent change over a 10-year period [40]. Moreover, in a subanalysis of the PREDyCES study in 401 oncology patients, 33.9% were at nutritional risk at hospital admission and only third of them had a nutritional intervention during their hospitalization [41]. These findings are consistent with the data in our study of 39% of patients at nutritional risk on admission and 48% of the whole population receiving nutritional support during hospitalization.

It should be noted that 20.7% of patients without nutritional risk on admission showed moderate/severe risk of malnutrition on discharge. Moreover, 47.4% of patients with severe nutritional risk remained at severe risk at discharge. These findings indicate that improvements in nutritional therapy during hospitalization are a crucial factor to prevent an increase in nutritional risk, particularly in the subgroup of patients with severe malnutrition on admission, as well as in those with small intestine, esophageal, and gastric cancer because these tumor sites were predictive factors of malnutrition. In addition, the fact that complications were significantly more frequent among patients with severe risk of malnutrition either on admission to the hospital or at the time of discharge further supports the need for meeting the nutritional needs of gastrointestinal surgical oncology patients. It is important to consider that, among patients who received nutritional therapy during hospitalization, 43.7% of patients continued to present moderate/severe risk of malnutrition and more than 50% with a MUST score ≥ 2 on admission maintained the same score at hospital discharge. Also, in patients with a MUST score ≥ 2 at discharged who had received nutritional therapy, 78% continued to have severe malnutrition. Therefore, clinicians should be especially aware of the nutritional needs of cancer patients at severe nutritional risk.

Despite extensive recognition that malnutrition is a critical predictor of toxicity and outcome in patients with cancer, physicians underestimate the influence of malnutrition-associated symptoms on the quality of care [30]. Lack of human resources, better nutrition training, and the creation of nutrition teams to routinely perform nutritional screening have been suggested to optimize management and improve efficacy during cancer treatments [42,43]. In a recent consensus nutritional approach for cancer patients in Spain, experts agreed that multidisciplinary action protocols that include nutritional and/or sarcopenia screening need to be developed in oncology clinics, and that nursing staff should routinely perform nutritional screening before starting cancer treatment [44].

## 5. Conclusions

This observational, multicenter, and exploratory study carried out in cancer patients with gastrointestinal tumors undergoing elective major abdominal surgery confirms that as much as 39% of patients were at risk of malnutrition on admission to hospital according to the MUST score, with 43.7% of patients who received nutritional therapy continuing to have moderate/severe nutritional risk on discharge. The potential relationship between nutritional risk at admission and significantly increased mortality rate at one year merits evaluation in further studies. However, based on the present results in a large group of gastrointestinal cancer patients undergoing elective surgery, there is an urgent need to improve nutritional risk screening before and after surgery, as well as to improve nutritional therapy during hospitalization.

## Figures and Tables

**Table 1 nutrients-15-03283-t001:** Salient characteristics of the study population on admission to the hospital.

Variables	Number (%)
Gender	
Man	291 (62.0)
Woman	178 (37.9)
Age, years, mean (SD)	68.2 (11.7)
Smoking habit	
Smoker	70 (14.9)
Ex-smoker	177 (37.7)
Never smoker	222 (47.3)
Weight, kg, mean (SD)	72.5 (13.9)
Weight loss in the previous 6 months	
<5%	307 (65.5)
5–30%	161 (34.3)
>30%	1 (0.2)
Height, m, mean (SD)	1.65 (0.10)
BMI, kg/m^2^, mean (SD)	26.7 (4.3)
<18.5	6 (1.3)
18.5–24.9	163 (34.7)
25–29.9	205 (43.7)
≥30	95 (20.3)
Comorbid conditions	
Cardiovascular disease (hypertension, heart disease, etc.)	271 (57.8)
Diabetes mellitus	110 (23.4)
Chronic obstructive lung disease (COPD)	51 (10.9)
Other	178 (37.9)
Charlson comorbidity index, mean (SD)	2.86 (1.81)
Barthel index score, mean (SD)	97.03 (9.4)
Total dependency	2 (0.4)
Severe dependency	4 (0.8)
Moderate dependency	48 (10.2)
Slight dependency	28 (6.0)
Totally independent	387 (82.5)
Oncologic treatment prior to admission	
Chemotherapy	108 (23)
Radiotherapy	72 (15.3)
Immunosuppressants for at least 3 months	6 (1.3)
Chronic corticosteroids (equivalent to 5 mg prednisone for at least 3 weeks)	4 (0.8)

Data expressed as frequencies and percentages unless otherwise stated.

**Table 2 nutrients-15-03283-t002:** Tumor site and surgical procedures in the study population of 469 patients.

Location of Primary Tumor	Number of Patients (%)
Colon	253 (53.9)
Rectum	111 (23.7)
Stomach	38 (8.1)
Pancreas	25 (5.3)
Esophagus	17 (3.6)
Small bowel	8 (1.7)
Liver	5 (1.1)
Biliary tree	4 (0.8)
Missing	8 (1.7)
Surgical procedures	
Colonic surgery	
Right hemicolectomy	128 (27.3)
Left hemicolectomy/sigmoidectomy	122 (26.0)
Rectal surgery	
Low anterior resection	87 (18.6)
Abdominoperineal resection	30 (6.4)
Total gastrectomy	40 (8.5)
Pancreatic surgery	28 (6.0)
Liver surgery	19 (4.1)
Esophageal surgery	15 (3.2)
Inclusion in an ERAS program	
Yes	193 (41.1)
No	276 (58.8)
Nutritional support at hospital admission	
Yes	158 (33.7)
No	611 (66.3)
Nutritional support during hospitalization	
Yes	231 (49.2)
No	237(50.53)
Missing	1 (0.2)
Length of ICU stay, days, mean (SD)	0.8 (3.9)
Length of hospital stay, days, mean (SD)	10.2 (9.8)

Data as frequencies and percentages in parenthesis unless otherwise stated.

**Table 3 nutrients-15-03283-t003:** Prevalence of malnutrition in 469 gastrointestinal surgical patients on hospital admission.

Nutritional Risk	MUST Score, *n* (%) (95% Confidence Interval)			GLIM Score *n* (%)
	MUST 0	MUST 1	MUST ≥ 2	
No risk	286 (61.0) (56.6–65.4)			
Moderate		84 (17.9) (14.4–21.4)		35 (35.3)
Severe			99 (21.1) (17.4–24.8)	64 (64.6)

**Table 4 nutrients-15-03283-t004:** Changes in the MUST score in gastrointestinal surgical oncology patients between hospital admission and discharge.

Hospital Discharge	Hospital Admission, *n* (%)			Total
	MUST 0	MUST 1	MUST ≥ 2	
MUST score 0	226 (79.3)	43 (51.2)	37 (38.9)	306 (66.0)
MUST score 1	47 (16.5)	33 (39.3)	13 (13.7)	93 (20.0)
MUST score ≥ 2	12 (4.2)	8 (9.5)	45 (47.4)	65 (14.0)
Total	285 (100)	84 (100)	95 (100)	464 (100) *

* Data of five patients at hospital discharge were missing.

**Table 5 nutrients-15-03283-t005:** Variables associated with the risk of malnutrition at hospital admission in the bivariate analysis.

Variables	*p* Value
Age	0.073
Civil status (single, married/partner, widow, separated/divorced)	0.111
Living conditions (alone, partner/family, institutionalized)	0.107
Body mass index	0.0001
Diabetes mellitus	0.074
Cardiovascular disease (hypertension, heart disease, etc.)	0.180
Barthel index	0.008
Esophageal cancer	0.009
Gastric cancer	0.015
Small bowel cancer	0.061
Colon cancer	0.010
Biliary tree cancer	0.159
Prediction of mortality attributable to comorbid disease in the short term (<3 years)	0.062
Prediction of mortality attributable to comorbid disease in prolonged follow-up (>5 years)	0.050

**Table 6 nutrients-15-03283-t006:** Predictive variables of malnutrition (MUST score ≥ 1) at hospital admission.

Variables	Wald χ^2^	*p* Value		Odds Ratio (95% CI)
Small bowel cancer	4.90	0.027	6.215 (1.23–31.33)	
Esophageal cancer	7.08	0.008	4.271 (1.47–12.44)	
Gastric cancer	7.34	0.007		2.567 (1.30–5.07)
Diabetes mellitus	1.62	0.031	1.623 (1.04–2.52)	

**Table 7 nutrients-15-03283-t007:** Variables associated with the risk of malnutrition at discharge from the hospital in the bivariate analysis.

Variables	*p* Value
Body mass index	0.0001
Barthel index on hospital admission	0.019
Esophageal cancer	0.005
Gastric cancer	0.017
Colon cancer	0.024
Primary liver cancer	0.171
Liver metastases	0.067
Pancreatic cancer	0.028
Digestive cytoreductive surgery	0.162
Esophagectomy	0.007
Gastrectomy	0.029
Non-anatomical liver resection	0.010
Cephalic duodenopancreatectomy	0.032
Prediction of mortality attributable to comorbid disease in the short term (<3 years)	0.123
Prediction of mortality attributable to comorbid disease in prolonged follow-up (>5 years)	0.017
Nutritional treatment during hospitalization	0.0001
Complications during hospitalization	0.0001
MUST score ≥ 1 on hospital admission	0.0001

**Table 8 nutrients-15-03283-t008:** Predictive variables of malnutrition (MUST score ≥ 1) at hospital discharge.

Variables	Wald χ^2^	*p* Value	Odds Ratio (95% CI)
MUST score ≥ 1 on admission	44.21	<0.0001	4.307 (2.80–6.62)
Complications, yes vs. no	13.40	0.0003	2.326 (1.48–3.24)
Nutritional treatment, yes vs. no	10.42	0.0012	2.079 (1.33–3.24)
Cytoreductive surgery, yes vs. no	4.852	0.0276	0.242 (0.069–0.85)

**Table 9 nutrients-15-03283-t009:** Relationship between complications and nutritional risk on admission to hospital and at discharge.

Nutritional Risk	Complications during Hospitalization *n* (%)		Total
	Yes	No	
Hospital admission			
MUST score 0	81 (55.5)	202 (63.9)	283 (61.2)
MUST score 1	20 (13.7)	64 (20.2)	84 (18.2)
MUST score ≥ 2	45 (30.8)	50 (15.8)	95 (20.6)
Total	146 (100)	316 (100)	462 (100)
Hospital discharge			
MUST score 0	75 (51.4)	229 (72.5)	304 (65.8)
MUST score 1	35 (24.0)	58 (18.3)	93 (20.1)
MUST score ≥ 2	36 (24.7)	29 (9.2)	85 (14.1)
Total	146 (100)	316 (100)	462 (100) *

* Missing data in seven patients because it was unknown whether they had complications or not as this item was not completed.

**Table 10 nutrients-15-03283-t010:** Changes in the MUST score in 231 patients who received nutritional therapy during their stay in the hospital.

Hospital Discharge	Hospital Admission, *n* (%)			Total
	MUST 0	MUST 1	MUST ≥ 2	
MUST score 0	87 (73.7)	20 (46.5)	23 (32.9)	130 (56.3)
MUST score 1	25 (21.2)	18 (41.9)	9 (12.9)	52 (22.5)
MUST score ≥ 2	6 (5.1)	5 (11.6)	38 (54.3)	49 (21.2)
Total	118 (100)	43 (100)	70 (100)	231 (100)

**Table 11 nutrients-15-03283-t011:** Changes in GLIM score in patients who received nutritional therapy during hospitalization.

Hospital Discharge	Hospital Admission, *n* (%)		Total
	Moderate Malnutrition	Severe Malnutrition	
Moderate malnutrition	7 (58.3)	1 (3.8)	8 (21.0)
Severe malnutrition	5 (41.7)	25 (96.1)	30 (78.9)
Total	12 (100)	26 (100)	38 (100)

## Data Availability

Data of the study are available from the corresponding author upon request.

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
