# Peer review of "The Prevalence and Impact of Nutritional Risk and Malnutrition in Gastrointestinal Surgical Oncology Patients: A Prospective, Observational, Multicenter, and Exploratory Study"

_nutrients, 2023, doi:10.3390/nu15143283_

Round 1

Reviewer 1 Report

Manuel Durán Poveda et al. submitted a comprehensive clinical reports on the relation between malnutrition and gastrointestinal surgical oncology patients. The authors described in detail about screening tool, eligible patients and their general characteristics and objectives that are anticipated to reach. More importantly, very detailed discussion on the report was involved. This manuscript is of high quality and can meet the requirement of special issue: New Perspectives for Cancer Patients’ Nutritional Support and Therapy. The authors can add the additional details mentioned below.

1. Abstract, line 33: The terminology “nutritional therapy” was mentioned multiple times in this manuscript and this therapy seems essential to rescue patients in malnutrition. But which kind of nutritional therapy are applied on these patients? It is an important part of this report and will be very illuminative to the potential readers of Nutrients. Hence it is strongly recommended for authors to add the detail of nutritional therapy.

2. Page 9, line 274: It is suggested to address the reason why 7 patients’ data is missing to avoid the possibility of subjectively data removal.

Here are some comments on format of the manuscript.

3. Page 2, line 56: different font format was applied here, are the authors intended to emphasize something here?

4. All the tables should be in three-line table format.

Author Response

Manuel Durán Poveda et al. submitted a comprehensive clinical reports on the relation between malnutrition and gastrointestinal surgical oncology patients. The authors described in detail about screening tool, eligible patients and their general characteristics and objectives that are anticipated to reach. More importantly, very detailed discussion on the report was involved. This manuscript is of high quality and can meet the requirement of special issue: New Perspectives for Cancer Patients’ Nutritional Support and Therapy. The authors can add the additional details mentioned below.

  • All authors would like to thank the Reviewer for positive and supportive comments regarding the scientific contribution of the study.

  1. Abstract, line 33: The terminology “nutritional therapy” was mentioned multiple times in this manuscript and this therapy seems essential to rescue patients in malnutrition. But which kind of nutritional therapy are applied on these patients? It is an important part of this report and will be very illuminative to the potential readers of Nutrients. Hence it is strongly recommended for authors to add the detail of nutritional therapy.
  • We have added this information in the Results section: “Of the 158 patients on nutritional support at the time of admission to the hospital, 154 (97.5%) received nutritional supplements.”
  • “A total of 231 patients (49.2%) received nutritional therapy during their stay in the hospital, including oral nutritional supplements in 57.3% of cases, total parenteral nutrition through central line in 38.7%, peripheral parenteral nutrition in 24.9%, and enteral nutrition through a nasojejunal feeding tube or jejunostomy in 4.9%.”

  1. Page 9, line 274: It is suggested to address the reason why 7 patients’ data is missing to avoid the possibility of subjectively data removal.
  • In the footnote of Table 9, we have clarified: “Missing data in 7 patients because it was unknown whether they had complications or not as this item was not completed.”

Here are some comments on format of the manuscript.

3. Page 2, line 56: different font format was applied here, are the authors intended to emphasize something here?

  •  

  1. All the tables should be in three-line table format.
  • Tables are adapted to the template of the Journal.

Reviewer 2 Report

The manuscript titled "Prevalence and Impact of Nutritional Risk and Malnutrition in Gastrointestinal Surgical Oncology Patients: A Prospective, Observational, Multicenter, and Exploratory Study" provides valuable insights into the prevalence of malnutrition and nutritional risk in gastrointestinal cancer patients undergoing major abdominal surgery. The study emphasizes the need for improved nutritional therapy and screening protocols to address this clinically relevant problem. The comprehensive analysis of predictors of malnutrition on admission and discharge enhances the understanding of managing these patients. Despite nutritional support during hospitalization, a significant proportion of patients remained at high risk of malnutrition. The manuscript is recommended for publication with minor revisions to address limitations and clarify certain sections. 

Firstly, in the Results section, it is important to clearly state the finding regarding patients with nutritional risk at admission experiencing a significantly increased mortality rate within one year after discharge. This information should be emphasized and reiterated in the Discussion section to underscore its significance. Additionally, the implications of this finding should be explicitly addressed in the Conclusion section to highlight its relevance and impact. By incorporating these revisions, the manuscript will effectively communicate the crucial findings to readers, enhancing their understanding and appreciation of the study's importance.

Author Response

The manuscript titled "Prevalence and Impact of Nutritional Risk and Malnutrition in Gastrointestinal Surgical Oncology Patients: A Prospective, Observational, Multicenter, and Exploratory Study" provides valuable insights into the prevalence of malnutrition and nutritional risk in gastrointestinal cancer patients undergoing major abdominal surgery. The study emphasizes the need for improved nutritional therapy and screening protocols to address this clinically relevant problem. The comprehensive analysis of predictors of malnutrition on admission and discharge enhances the understanding of managing these patients. Despite nutritional support during hospitalization, a significant proportion of patients remained at high risk of malnutrition. The manuscript is recommended for publication with minor revisions to address limitations and clarify certain sections. 

  • We sincerely appreciate your comments regarding the scientific interest of the study.

Firstly, in the Results section, it is important to clearly state the finding regarding patients with nutritional risk at admission experiencing a significantly increased mortality rate within one year after discharge. This information should be emphasized and reiterated in the Discussion section to underscore its significance. Additionally, the implications of this finding should be explicitly addressed in the Conclusion section to highlight its relevance and impact. By incorporating these revisions, the manuscript will effectively communicate the crucial findings to readers, enhancing their understanding and appreciation of the study's importance.

  • Short (< 3 years) and long-term (> 5 years) outcomes refer to prediction of death attributable to comorbid disease based on the Charlson comorbidity index (reference #33). Throughout the text we have clarified the meaning of prediction of mortality attributable to comorbid disease.
  • The relationship between patients with nutritional risk at admission and significantly increased mortality rate at one year was not evaluated. We have added this point in the Conclusions: “The potential relationship between nutritional risk at admission and significantly increased mortality rate at one year merits to be evaluated in further studies.”
